# ePSICONUT: An e-Health Programme to Improve Emotional Health and Lifestyle in University Students

**DOI:** 10.3390/ijerph19159253

**Published:** 2022-07-28

**Authors:** Luisa Marilia Cantisano, Rocio Gonzalez-Soltero, Ascensión Blanco-Fernández, Noelia Belando-Pedreño

**Affiliations:** 1School of Psychology, Pontificia Universidad Católica Madre y Maestra (PUCMM), Santiago De Los Caballeros 51000, Dominican Republic; luisacantisano@gmail.com; 2Department of Medicine, Faculty of Biomedical and Health Sciences, Universidad Europea de Madrid, 28670 Villaviciosa de Odón, Spain; ascension.blanco@universidadeuropea.es; 3Faculty of Sports Sciences, Department of Physical Activity and Sport, Universidad Europea de Madrid, 28670 Villaviciosa de Odón, Spain

**Keywords:** eHealth, mHealth, lifestyle, subjective well-being, healthy lifestyle habits

## Abstract

The use of information and communication technologies in the health field is known as eHealth. Nowadays, the application of technological and digital tools for maintaining/improving physical and mental health is experiencing an exponential boom. These tools have been perceived as a powerful support for face-to-face therapies and lifestyle changes. Nevertheless, there is not enough scientific research that analyses the impact and consequences of eHealth interventions. More studies are needed to validate its application. Therefore, the aim of this study was to evaluate the impact of eHealth tools in a programme called ePSICONUT. This programme was created to promote healthy lifestyle habits in university students. The sample consisted of 16 university students from the Dominican Republic aged 18–24 years ($\bar{x}\;$ = 20.69; s = 1.74). ePSICONUT was developed in 12 weeks and its impact was analyzed by comparing the initial and the final psychological and lifestyle tests results, which were completed online by the participants. Results reported that the professionally supervised use of eHealth tools was associated with better psychological well-being, lees anxiety and depression, and better lifestyle habits (such as diet quality), even in stressful and changing situations such as the COVID-19 pandemic circumstances. However, more studies are needed to validate and promote the use of eHealth-based intervention programmes.

## 1. Introduction

A chronic unhealthy lifestyle and a poor use of emotional regulation techniques could damage mental and physical health [1]. Research has identified that this situation is more perceived in societies where people have to dedicate long hours to work and family care, making it difficult to maintain proper lifestyle and self-care activities [2]. Epidemiological evidence suggests that the integration of the use of technological and digital tools could significantly improve healthcare services. Results from different studies reported how this is a potential alternative for health promotion, cost reduction, and prevention of physical and mental illnesses [3,4]. The use of information and communication technologies (ICT) in health field is called eHealth [3].

After the COVID-19 pandemic, the world turned its focus even more towards three aspects that have been in the spotlight for years: lifestyle (healthy eating and physical exercise), emotional well-being, and the use of technology and digital tools to improve them [5,6]. Health professionals, as well as health service users and the general population, have been using emails, SMS, electronic records, educational videos, telemedicine, online therapies, and mobile applications to promote mental health and a healthy lifestyle [4,5,7,8].

Among the digital resources most commonly used in physical activity and healthy eating fields are: mobile applications, self-reporting of behaviours in favour of a healthy lifestyle through digital tools, mobile devices such as podometers, online platforms that facilitates the communication between patients and health professionals (via smartphones or computers), and reminders and reinforcements through text messages or social networks to promote a healthy lifestyle or to improve the adherence to a nutritional treatment [8,9,10,11,12,13,14]. 

Moreover, there are apps that has been designed to reduce stress and anxiety [15]. Less studied but also promising is the use of eHealth to increase psychological well-being by tools related to positive psychology and third-generation therapies, such as mindfulness, specifically in the population with unhealthy lifestyle habits [16]. However, to effectively apply eHealth, the scientific validation of the different available digital tools must be carefully tested and evaluated. Nowadays, there are very few scientific studies that support its validity [12,15]. Moreover, some studies state that the use of eHealth tools should be supervised by a facilitator or health professional that generates social interaction and support [14]. It could be counterproductive for people’s health, as well as ineffective, to access and use these digital platforms without an expert supervision [9].

Unfortunately, most of the mobile and digital applications available in the market, promoted as alternatives to optimise lifestyle and mental health, have not been properly studied, especially those available for the Spanish-speaking population [17]. Because of the lack of scientific evidence and professional support, many of the ones related to lifestyle promote obsessions, pressures, and discomfort in subjects, such as calorie counting applications [17]. This limitation and the lack of standardised protocols and guidelines on how to use them properly do not allow health professionals to recommend their use [18,19,20].

The main objective of the present study was to evaluate the impact of the eHealth tools to improve lifestyle (physical activity and diet), as well as emotional well-being. To do so, a digital programme called ePSICONUT was designed by researchers and health professionals. It was applied in a group of university students from Dominican Republic. The hypothesis is that this online programme would improve the diet quality, physical exercise, and emotional state of the participants.

## 2. Materials and Methods

### 2.1. Study Design

ePSICONUT was based on a quasi-experimental study, with a pre-post design. It used a quantitative methodology. Conclusions of the programme’s impact come from intrasubject measures: a comparison between the answers that they give to a group of questionnaires before and after the intervention [21]. These questionnaires evaluated the variables under study: lifestyle (e.g., *overall lifestyle, diet quality* and *physical exercise*) and the ones related to the *psychological status* of the participants (e.g., *subjective psychological well-being, anxiety,* and *depression*).

### 2.2. Participants

The final sample of the study consisted of 16 Psychology and Nutrition and Dietetics students from Pontificia Universidad Católica Madre y Maestra (Santiago, Dominican Republic) aged between 18 and 24 years ($\bar{x}$ = 20.69; s = 1.74). The sample selection was made through considering the age range of undergraduate students who best use digital tools according to the literature, which is from 18 to 45 [22,23]. The sampling technique employed was a non-probability sampling by convenience and accessibility [24]. Participants were volunteers and met the inclusion criteria (Table 1). Figure 1 shows how the sample evolved from the beginning to the end of the study because of the criteria described in Table 1. It is important to highlight that the 16 participants that completed all the steps of the programme were women, though it was not exclusive for a female population.

This study has been designed and applied considering all the bioethical principles established by the Belmont Report [25] and the Declaration of Helsinki [26]: principles of autonomy beneficence, justice, and non-maleficence. This research was approved by the Research Ethics Committee of the European University of Madrid and, subsequently, ratified by the Bioethics Committee of the Faculty of Health Sciences (COBE-FACS) of the PUCMM, Santiago, Dominican Republic (ID for both institutions: CIPI/19/148).

### 2.3. Instruments

Digital instruments and psychometric questionnaires were used to determine the impact of the ePSICONUT programme, being classified as: (a) promotion of the programme tools; (b) test batteries applied at the beginning and at the end of the programme; (c) assessment of psychological state and lifestyle.

#### 2.3.1. Promotion of ePSICONUT Tools

For the recruitment of participants, the program was promoted though: “stories” published via Instagram, the institutional email of PUCMM, and WhatsApp groups of Nutrition and Dietetics and Psychology students from PUCMM. Participants had to complete the initial tests battery sent on 19 November 2020. To choose who was eligible to participate in the ePSICONUT programme, the responses of the subjects were reviewed by the researchers and were compared with the criteria listed in Table 1.

##### Test Batteries Applied at the Beginning and at the End of ePSICONUT

At the beginning, two questionnaires were created ad hoc in Google Forms to test the sample before the intervention. The questionnaires were: *A General Health Questionnaire*. Consisting of 13 questions, aimed to evaluate the presence/diagnosis of physical and/or mental pathologies, present or past, as well as risk and protective factors for health (such as sleep, smoking, coffee intake, and meditation).*A Digital Health Questionnaire*. It assesses the use and perception of participants about digital tools for the improvement of lifestyle and psychological well-being. Many questions of this instrument were based on a survey called *Encuesta de usuarios 2018 sobre sanidad digital de España [2018 Spanish eHealth User survey]* [27]. In the initial questionnaires’ battery, this section had just 11 questions to evaluate these topics in general. In the final questionnaires’ battery, this part consisted of 24 questions: on the one hand, the same questions that included the initial test; on the other hand, questions that were focused on evaluating the subjects’ perception about the digital tools specifically used in ePSICONUT (Headspace, Insight Timer, Fabulous, YouTube channel, WhatsApp group, e-mail, and Excel sheets to perform some of the programme’s tasks/activities). Furthermore, this last questionnaire included an ad hoc Satisfaction Survey to evaluate the participants’ appreciation about the following aspects of the programme: organisation, quality of the follow-up offered, quality of the information provided, self-perception of the ePSICONUT impact on the improvement of their eating habits, physical exercise, and psychological well-being.

##### Instruments Used to Evaluate the Psychological State and Lifestyle of the Participants

*Beck Depression Inventory-II (BDI-II).* A 21-item multiple-choice questionnaire that measures the severity of current depression level in adults and adolescents [28]. It describes the emotional, physiological, and cognitive symptoms of depression. The total score ranges from 0 to 63 as follows: minimal depression from 0 to 13, mild from 14 to 19, moderate from 20 to 28, and severe from 29 or more. Its subscales and the general scale show moderate and high reliability when evaluated in the Dominican population, ω = 0.89. Similarly, adequate external validity was demonstrated by observing the adequate discrimination of the scale between the Dominican clinical population (with depression) and the general population (without depression) [29,30].

*State-Trait Anxiety Inventory (STAI).* A self-report used to assess the level of trait and state anxiety in the subject [31]. It has been validated in the Dominican population [32]. Each subscale consists of 20 items. The trait subscale is answered with one of the following options: never (0), almost never (1), sometimes (2), or often (3). The state item is answered with: not at all (0), somewhat (1), quite a lot (2), or very much (3). The scores that the subject can obtain in each subscale range from 0 to 60, and can be categorised as low or none, moderate, or high anxiety. This questionnaire has shown good internal consistency, α = 0.90 and 0.93, ω = 0.71 and 0.65, for state and trait anxiety, respectively.

*Dimension of Subjective Psychological Well-Being Subscale of the Psychological Well-Being Scale (EBP).* The EBP is a psychometric test created by Sánchez-Cánovas [33], which consists of 65 items that are scored on a Likert-type scale ranging from 1 to 5. This scale is composed of four subscales: subjective psychological well-being (SPWB), material well-being, occupational well-being, and relationships with partners. In ePSICONUT, only the first subscale was applied. The SPWB is related to the subjects’ perception of happiness in their lives, as well as their positive and negative feelings. It also assesses the individual’s ability to overcome the different stages of the life cycle, asking questions related to adolescence, youth, middle age, and old age. It is made up of 30 items that those evaluated must score with the following options: 1 = never or almost never; 2 = sometimes; 3 = quite often; 4 = almost always; 5 = always. The maximum score that can be obtained on the EBPS is 150 The reliability presented was ω = 0.96.

*University of Rhode Island Change Assessment Scale (URICA)*. This scale measures the stage of the change process in which a person is in: precontemplation, contemplation, preparation/action. It was originally created by McConnaughy in 1983 [34]. The version adapted to the Spanish context by Gómez-Peña et al. was used [35]. This scale consists of 32 items that evaluate the stage of motivation to change and commitment to treatment in which the patient is in. Each of the items is scored on a 5-point Likert-type scale: 1 = strongly disagree; 2 = disagree; 3 = undecided; 4 = agree; 5 = strongly agree.

*Healthy Lifestyle Scale for College Students*. This brief scale is a Mexican adaptation of the Healthy Lifestyle Scale for University Students [36]. This instrument basically evaluates four lifestyle dimensions that have been positively or negatively related to chronic noncommunicable diseases: substance use; appreciation for life; interpersonal relationships; and eating, studying, and sleeping patterns. In total, it consists of 14 items scored on a 5-point Likert scale (1 = never; 2 = rarely; 3 = sometimes; 4 = usually; 5 = always). The internal consistency was ω = 0.81. 

*Overall diet quality index.* The index used during ePSICONUT was created by Ratner et al. [37] on the basis of recommendations provided by the dietary guidelines of Latin American countries, such as the Dominican Republic. It evaluates how often people consume: (1) “healthy foods”; (2) “unhealthy foods”, including cakes, cookies, sweets (as a group), sugar, sugar-containing drinks, and fried foods; (3) “meals”, indicating whether breakfast, lunch/lunch and dinner are consumed. Participants are asked to rank how often they consume each of the above items using a 6-choice Likert scale. The quality of the diet is classified as follows: 90–120 points = healthy; 60–89 points = needs changes; <60 points = unhealthy.

*International Physical Activity Questionnaire (IPAQ).* The short version of the self-administered validated questionnaire in the Hispanic context by Mantilla Toloza and Gómez-Conesa [38] was used. It allows for the observation and monitoring of the physical activity of the population aged 18 to 65 years. The internal consistency was ω = 0.88.

### 2.4. Procedure and Design of the Intervention

All those participants who answered the initial test battery received either a welcome email or an email with the reasons why they could not participate in the programme. In the email of those admitted, a link was sent with the survey “Welcome to ePSICONUT: change your habits, improve your life!”. In this survey, their WhatsApp number was asked to create a group with all the participants of ePSICONUT. The programme took place for 12 weeks. The intervention programme is described in Table 2 and represented in Figure 2.

### 2.5. Data Analysis

Descriptive statistics and pre-post statistically differences were calculated for the variables under study: the ones related to *lifestyle* (e.g., *diet quality and physical exercise*) and the ones related to the *psychological status* of the participants (e.g., *subjective psychological well-being, anxiety,* and *depression*). The normality test was applied to the variables using the Shapiro–Wilk statistics [39]. On the basis of a confidence level of 95% and an α error of 5% (0.05), the null hypothesis (that the distribution was normal) was rejected, *p*-value ≤ 0.05. The reliability of the psychometric questionnaires was calculated through Cronbach’s Alpha and McDonald’s Omega statistic [40]. 

To assess the impact of PSICONUT pre-intervention and post-intervention, the Wilcoxon signed-rank test and Student’s t-test for related samples were applied. The Wilcoxon non-parametric test was used to compare data that did not meet the normality criterion [41]. In the case of variables that presented a normal distribution, Student’s *t*-test was used, which works very well for *n* ≤ 30 (as in this study) [42]. Effect sizes were calculated to quantify the magnitude of the differences between the pre- and post-intervention phase of the ePSCONUT programme. Cohen suggests that d ± 0.2–0.5 represents a “small” effect size, ±0.5–0.8 a “medium” effect size, and >±0.80 a “large” effect size [43]. Finally, statistical power was calculated using post hoc analysis difference between two means and assuming the minimum power level of 0.80 following Ellis [44]. In the case of effect size in variables where the Wicolxon rank test is applied, biserial rank correlation (*rbis)* is used [45]. Statistically significant differences were considered as *p* ≤ 0.05. Analyses were run with the statistical package SPSS 25.0 (*Statistical Package for Social Sciences.* New York, NY, USA), Jamovi 2.5, and G*Power 3.1 (Axel Buchner, Düsseldorf, Germany) (the last one for the calculation of statistical power).

## 3. Results

### 3.1. Impact of ePSICONUT on Global Diet Quality Index

The differences before and after ePSICONUT in intake of certain foods, breakfasts, lunches, and dinners, as well as the overall quality of the diet, will be presented below. Differences between variables with normal distribution were evaluated with Student’s t-test (see Table 3), while the rest were evaluated with the Wilcoxon test (see Table 4). To interpret the results, it is necessary to remember that the mean and median values presented, correspondingly, are based on the scores obtained by the participants when answering the Global Diet Quality Index. This means that the higher the score (hence, the mean or median) obtained after the sum of the three areas evaluated by this index, the better the diet. Consequently, on the one hand, the higher the score for “healthy foods” (vegetables, fruits, milk or dairy products, legumes, and fish) and “meals”, better the diet; on the other hand, the higher the score for “unhealthy foods” (cakes, cookies and sweets, sugary drinks, sugar, and fried foods), the poorer the diet quality. 

As can be seen in Table 3 and Table 4, the overall diet quality of the participants improved significantly (*p* < 0.05) after participating in the ePSICONUT programme. In general, there was a greater increase in the consumption of “healthy foods”, as well as a decrease in “unhealthy foods”. Specifically, there was a statistically significant increase in the consumption of vegetables (recall that such increase is evidenced by a higher score on this index), as well as a statistically significant decrease in the consumption of sugar-sweetened beverages (recall that such decrease is evidenced by a higher score on this index).

### 3.2. Impact of ePSICONUT on Physical Exercise

To evaluate the impact of ePSICONUT on physical exercise, the difference in the responses given in the *IPAQ* before and after participating in the programme was evaluated. There was a slight increase in the level of physical activity after participating in the ePSICONUT programme (see Table 5). However, this was not statistically significant (*p* > 0.05). 

### 3.3. Impact of ePSICONUT on Overall Lifestyle

In addition, to evaluate the participants’ diet and physical exercise, ePSICONUT examined other lifestyle habits of the students, such as rest, leisure, relationships with their peers and with themselves, and their organisation in their studies, among other aspects included in the *Healthy Lifestyle Scale* for University Students.

When comparing, through a Student’s *t*-test, the means of the scores obtained before ($\bar{x}$ = 52.69) and after ($\bar{x}$ = 57.13) the ePSICONUT programme, a statistically significant difference was observed (t [gl = 123] = 3.02, *p* = 0.009). The effect size value was *d* = 0.076 and statistical power of 1 − *ß* = 0.95. This is evidence that the overall lifestyle of the students improved after the programme.

### 3.4. Impact of ePSICONUT on Subjective Well-Being and Motivation to Change

Table 6 presents the comparisons, using the Wilcoxon signed-rank test, of before and after ePSICONUT in the following aspects: subjective well-being and levels of depression. Subjective well-being increased and depression levels decreased, in a statistically significant way, after applying the programme (*p* < 0.05). On the other hand, the differences between the means of the levels of anxiety and motivation for change, before and after ePSICONUT, were evaluated by means of a Student’s *t*-test (see Table 7). It was observed that anxiety decreased significantly, including trait anxiety (*p* < 0.05). As for motivation to change, no statistically significant differences were found between before and after the programme (*p* > 0.05).

### 3.5. Program Impact

Table 8 shows descriptive statistics of the evaluation made by the participants on the quality criteria of the programme. It is observed that the majority of the participants rated the ePSICONUT programme as excellent. Likewise, the majority stated that the programme had a very high impact on their eating habits and subjective well-being. The aspect they rated as having the least impact (although tending to have a high impact) was physical exercise.

## 4. Discussion

ePSICONUT is an online interventional programme specifically designed for this research. The main objective of the study was to assess the impact of eHealth on the lifestyle and psychological well-being of a group of university students. It also tried to be a support to fill the gap that exists today around the validity and effectiveness of the use of ICT interventions in the health field, specifically in healthy lifestyle and mental health promotion [12,15,20]. We included digital tools that had demonstrated good quality and effectiveness in optimising life habits and emotional well-being. Among these, we can point out: Headspace (Santa Monica, CA, USA) [46,47], Insight Timer (San Francisco, CA, USA), Fabulous (Paris, France), WhatsApp (Meta Platforms, Cambridge, MA, USA) [48], YouTube (San Mateo, CA, USA) [49], Zoom (San Jose, CA, USA) [50], and email [51].

The results derived from this intervention indicate that students significantly improved their eating habits at the end of the programme. Throughout the implementation of the programme, an online nutritional education strategy was used to improve the diet quality: nutritional recommendations through informative capsules posted on YouTube and sent by email; encouraging the use of “El Nutriplato” [52] as a guiding tool to organise lunches and dinners; promoting the use of some techniques from the app called Fabulous (such as goals of drinking water, eating a large breakfast, and eating more fruits and vegetables). As a result, students tended to increase healthy foods intake (e.g., vegetables and greens) and to decrease the consumption of low nutritional quality food, such as sugary drinks. This result supports the promotion of the use of eHealth tools to optimise dietary quality, as well as the findings of other eHealth nutrition programmes that observed a significant improvement in diet and overall lifestyle in clinical [49] and non-clinical populations [53,54] after participating in programmes that facilitated the communication and education process between patient and health professional via WhatsApp, Facebook, or SMS. ePSICONUT also used WhatsApp as one of its main working tools.

ePSICONUT was designed and developed under a group modality, as well as online. It has been shown that focus-group interventions can be highly beneficial in promoting motivation and engagement in habit consolidation and changes, as well as in general self-care and decreasing negative emotions [48,55,56,57]. This same need for closeness, support, and modelling may also have had an impact on why all synchronous media used in the programme tended to be the most preferred and used by participants. Although authors such as Lie et al. [58] have found high efficacy in asynchronous eHealth media, this kind of telehealth may also represent a challenge in communication between healthcare professionals and service users [59]. 

In terms of the exercise practice, although constant reminders were given and motivation to exercise was promoted via WhatsApp, no statistically significant differences were evident between the level and frequency of physical exercise performed before and after the programme. This information on how to practice physical exercise properly was given by an asynchronous means (videos posted on YouTube). It is likely that synchronous sessions could be implemented to have better results.

Furthermore, in the psychological aspects, a statistically significant increase in subjective well-being was observed, as well as a statistically significant decrease in anxiety and depressive symptoms. On the basis of what has been explained and observed by other authors [51,60,61,62,63,64], as well as by the ePSICONUT participants themselves, psychological techniques and strategies, such as mindful eating, mindfulness, gratitude tasks, body acceptance exercises, and the same group support (all supported by the use of eHealth media), developed and practiced during the programme, could play an important role in the improvement of the psychological status that was observed after applying ePSICONUT in the participants. 

Despite all these positive findings, it is important to clarify that this study has a limitation: the sample size. As can be seen in Table 1, 40 was the number of students that accepted to participate in this online programme. Only 10 dropped out of the study at some point. This means that 75% of the participants who started the program remained active, in one way or another, until its end. Thus, the dropout rate was not as large when compared to other intervention studies of 2 months or more [54]. However, this study was a pre–post intervention whose impact on lifestyle and psychological well-being pretended to be measured by intrasubject answers. Only 16 participants completed the final test battery, which was needed in order to be compared with the initial test battery because of the nature of this study. Although online questionnaires and surveys tend to be resources that are easy to manage and broadcast, Nayak and Narayan [65] confirmed, on the basis of their own studies and on other researchers’ experience, that this class of methods has a great disadvantage: the response rates by the subjects are usually much lower than those of offline and face-to-face questionnaires. This disadvantage and the fact that the online questionnaire for this program was quite long could have been key factors in our obtaining of only 16 complete responses. Although Hernández Sampieri et al. [66] state that 15 participants are the minimum for quasi-experimental studies, one potential purpose for the future is to conduct further studies based on ePSICONUT preliminary results but with larger samples and a control group. They could reinforce more the use of eHealth to promote a better lifestyle and psychological status.

Another limitation of this research is that no control group was used. In this sense, in quasi-experimental design studies, there is no randomisation of subjects to treatment and control groups, or there is no control group per se [67]. In the same line of research, García-Solano et al. [68] carried out a similar study in working adult population, without a control group, in which participants showed a behaviour change towards a healthy lifestyle.

## 5. Conclusions

University students who completed ePSICONUT (a psychonutritional programme fully supported by eHealth tools) showed a significant improvement in their lifestyle at the end of the programme. Although there was no statistically significant difference in exercise before and after the programme, other habits, such as healthy eating, were optimised. Consumption of high nutritional quality foods increased, and consumption of lower nutritional quality foods decreased. There was a significant and favourable change in the consumption of vegetables and greens, as well as a significant decrease in sugary drinks. 

The mental health of the ePSICONUT participants also increased significantly at the end of the programme. Students showed an increase in their subjective well-being, as well as a decrease in negative emotions such as anxiety and depressive symptoms.

## Figures and Tables

**Figure 1 ijerph-19-09253-f001:**
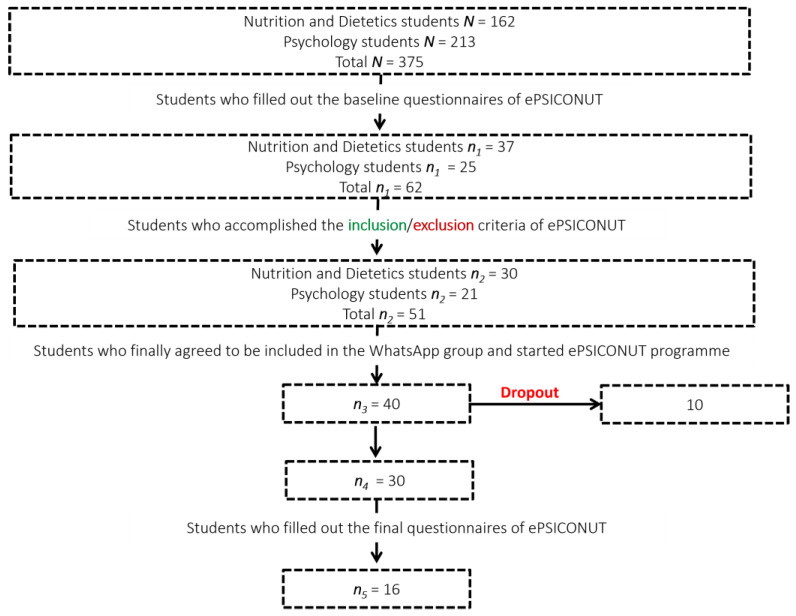
Evolution of the sample throughout the development of ePSICONUT programme.

**Figure 2 ijerph-19-09253-f002:**
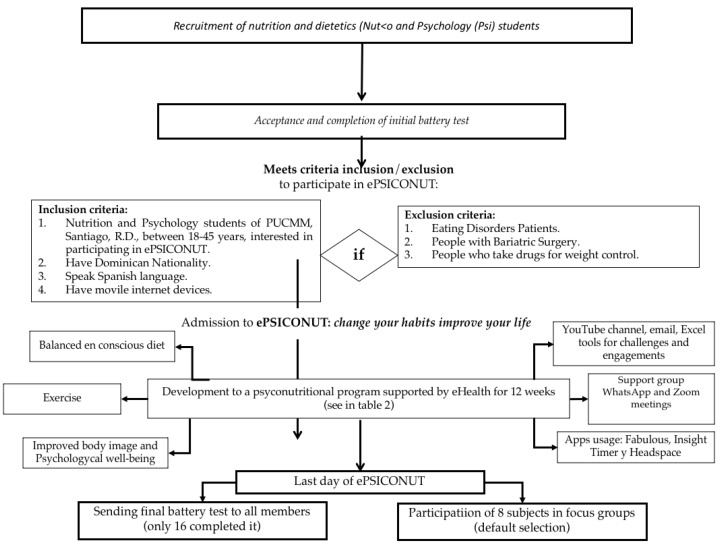
Diagram to summarise the experimental protocol of ePSICONUT.

**Table 1 ijerph-19-09253-t001:** Inclusion and Exclusion Criteria of ePSICONUT programme.

*Inclusion Criteria*	*Exclusion Criteria*
Students of Nutrition and Dietetics or Psychology from PUCMM, Santiago, D.R., between 18 and 45 years old, who agree to participate in ePSICONUT.	Dropout from the psychonutritional intervention programme.
Have Dominican nationality.	People with an eating disorder.
Speak Spanish.	People undergoing bariatric surgery.
Have mobile devices with Internet services.	People taking drugs related to weight control.

**Table 2 ijerph-19-09253-t002:** Activities carried out during the 12 weeks of the ePSICONUT programme.

*Weeks*	*Activities*
** *Week 1* **	*Creation of WhatsApp group and welcome and sending mail with video on YouTube channel.*
** *Week 2* **	*Reminders of commitments and clarification of doubts by mail and WhatsApp.*
** *Week 3* **	*Reminders of commitments and challenges via WhatsApp.*
** *Week 4* **	*Continuation of messages by WhatsApp to take care of food and follow the exercise to strengthen the psychological well-being.*
** *Week 5* **	*Meeting via Zoom to address the balance between self-care and study habits. Fabulous was introduced as the mobile application to create, strengthen, and track healthy habits.*
** *Week 6* **	*Informative capsule posted on the YouTube channel to promote the organisation and optimisation of food quality; provide recipes/options for healthy breakfasts, lunches, dinners, and snacks; introduce the Nestlé Nutriplato.*
** *Week 7* **	*Videos posted on YouTube channel to promote physical exercise at home and in nature (according to the limitations by COVID-19 pandemia situation).*
** *Week 8* **	*Meeting via Zoom to introduce mindfulness practice (led by the principal investigator and a certified mindfulness practitioner). The use of Headspace was encouraged to practice mindfulness on a daily basis thereafter.*
** *Week 9* **	*Video via YouTube channel to introduce mindful eating. Insight Timer is presented as a mobile application to support the practice of mindful eating.*
** *Week 10* **	*Sending an informative document: what it is, how to exercise, and optimising acceptance.*
** *Week 11* **	*Constant communication via WhatsApp, stimulating: conscious eating, physical exercise, body image acceptance. * *Meeting via Zoom to practice conscious eating at lunchtime.*
** *Week 12* **	*Sending of final test battery.* *Closing meeting via Zoom with a focus group of eight people, in order to make a qualitative analysis of the effectiveness, limitations, and areas for improvement of the ePSICONUT programme.*

**Table 3 ijerph-19-09253-t003:** Student’s *t*-test applied to food parameters evaluated in ePSICONUT.

Variables	$\bar{x}1$	$\bar{x}$ *2*	Sig. *t* Student	*d*	1 − *ß*
Healthy food					
Dairy products	5.44	6.31	0.371		
Total ICGD score	74.31	87.38	0.002 *	−0.92	0.93

Note. $\bar{x}$**1** = mean before starting the ePSICONUT programme; $\bar{x}$*2* = mean at the end of the ePSICONUT programme. GDI = Global Diet Quality Index. * *p* ≤ 0.05.

**Table 4 ijerph-19-09253-t004:** Wilcoxon test applied to food parameters evaluated in ePSICONUT.

Variables	$\tilde{x}$1	$\tilde{x}$2	Sig. Wilcoxon	*r_bis_*	1 − *ß*
Healthy foods					
Vegetables	3.75	7.50	0.005 *	−0.91	0.96
Fruits	1.75	7.50	0.063		
Legumes	10.00	10.00	0.317		
Fish	4.25	7.50	0.114		
“Unhealthy” foods					
Cakes, cookies, and sweets	5.00	6.25	0.258		
Sugary drinks	3.75	7.50	0.042 *	−0.64	0.58
Sugar	7.50	1.00	0.064		
Fried foods	7.50	7.50	0.885		
Meals					
Breakfasts	10.00	10.00	1.000		
Lunches	10.00	10.00	0.317		
Dinners	10.00	10.00	0.655		

Note. $\tilde{x}$1 = median before starting the ePSICONUT programme; $\tilde{x}$2 = median at the end of the ePSICONUT programme. * *p* ≤ 0.05

**Table 5 ijerph-19-09253-t005:** Wilcoxon test applied to physical exercise parameters evaluated in ePSICONUT.

Variables	$\tilde{x}$1	$\tilde{x}$2	Sig. Wilcoxon
MET	214.50	394.50	0.221
Weekly physical activity time	65.00	80.00	0.363

Note. $\tilde{x}$1 = median before starting the ePSICONUT programme; $\tilde{x}$2 = median at the end of the ePSICONUT programme; MET = metabolic equivalents.

**Table 6 ijerph-19-09253-t006:** Wilcoxon test applied to subjective psychological well-being and depression levels assessed in ePSICONUT.

Variables	$\tilde{x}$1	$\tilde{x}$2	Sig. Wilcoxon	*r_bis_*	1 − *ß*
BDI-II	13.00	3.50	0.001 *	0.97	0.99
EBPS	104.50	120.00	0.021 *	0.65	0.73

Note. $\tilde{x}$1 = median before starting the ePSICONUT programme; $\tilde{x}$2 = median at the end of the ePSICONUT programme. BDI-II = *Beck Depression Inventory-II*; EBPS = *Subjective Psychological Well-Being Subscale of the Psychological Well-Being Scale*. * *p* ≤ 0.05.

**Table 7 ijerph-19-09253-t007:** Student’s t-test applied to anxiety parameters evaluated in ePSICONUT.

Variables	$\bar{x}$1	$\bar{x}$2	Sig. t Student	*d*	1 − *ß*
STAI-S	24.63	15.56	0.002 *	0.94	0.93
STAI-T	25.00	18.44	0.017 *	0.67	0.71

Note. $\bar{x}$1 = mean before starting the ePSICONUT programme; $\bar{x}$2 = mean at the end of the ePSICONUT programme. STAI-S = State Anxiety; STAI-T = Trait Anxiety. * *p* ≤ 0.05.

**Table 8 ijerph-19-09253-t008:** Quantitative results of the ePSICONUT Programme Satisfaction Survey.

Criteria for Satisfaction with ePSICONUT	$\bar{x}$	$\tilde{x}$	*s*
Quality criteria			
Quality of the follow-up provided	4.88	5.00	0.34
Level of organisation	4.94	5.00	0.25
Quality of information provided	5.00	5.00	0.00
Evaluation of subjective impact			
Impact on physical exercise	3.94	4.00	1.24
Impact on psychological well-being	4.38	4.50	0.72
Impact on diet	4.50	5.00	0.73

Note. $\bar{x\;}$ = mean; $\tilde{x}\;$= median; *s* = standard deviation.

## Data Availability

Data will be available upon request. Data are not deposited in any public available database.

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
