# Peer review of "ePSICONUT: An e-Health Programme to Improve Emotional Health and Lifestyle in University Students"

_ijerph, 2022, doi:10.3390/ijerph19159253_

Round 1
Reviewer 1 Report
Expertise d’article
I received the paper entitled ‘ePSICONUT- e-health programme for the improvement of life-2 style and emotional health in the university population’ for expertise. After reading the latter, I have made some remarks that may help the authors to improve their work.
Overall, I found it difficult to understand concretely what the authors were aiming for as there are too many variables measured and I do not see the links between the different variables. The number of participants is very small, and I wonder why the authors did not use more participants. Furthermore, the authors chose to measure only trait anxiety (line 170). I wonder why they did not measure state anxiety. The experimental protocol does not include a control group, yet the authors claim that their intervention influences their dependent variables. I think the authors need to be tempered in their conclusion. As the questionnaires are self-reported, how do the authors check the veracity of the participants' answers. The choice of questionnaires used should also be justified. In sum, I think the study is not sufficiently standardised and lacks sufficient precision for readers to understand the significance of the study, although it is interesting.
Line 110 : Study or studio
Table 1 : People with ED. The author could define ED
Lines 110-116: The acceptance number of the ethics committee must be specified
Lines 140-158 : Why digital health questionnaire is different between pre and post intervention??? it's not standardised at all?
Line 147: Headspa-ce or Haedspace
Lines 179-191 : The author could define EBP, SPSS, SPBS? EBPS. Acronyms should be checked throughout the manuscript.
Lines 224 – 235 : I suggest that the authors make a diagram to summarise the experimental protocol.
Line 248 : The authors say that if the data are normally distributed, they use student's t. And if not, what do they do? It is in this section that they should specify it and not in another section as the authors have done.
Lines 25-35-94-95-122-125-268 and 252-455-457-…..463: choose between 'program' and 'programme' and harmonise throughout the document.
Table 3 and 5: The mixed English and Spanish authors. Care must be taken
Lines 290-302 : What do the observed correlations mean?
Table 8: There is no star in the table, yet the authors have put a star at the bottom of the table to indicate a significant effect.
Author Response
We are grateful for all the considerations made for the improvement of this manuscript. We respond to each of them below.
General comments
We thank the reviewers for their feedback, which allows the authors of this manuscript to improve the application of the scientific method and scientific writing.
In general, we have modified the wording of the introduction to better relate the variables under study. In the Methodology section, the instruments used to assess the dependent variables and the protocol are explained in greater detail.
In Data analysis, we have corrected conceptual errors about the non-parametric tests used.
In Results, we tried to improve the explanation of the instruments used and the statistically significant differences found.
Finally, Discussion has been readjusted and the explanations of the limitations of the study (sample size, non-use of a control group, etc.) have been expanded.
Response to the reviewer's considerations 1
Comments to the Author
Overall, I found it difficult to understand concretely what the authors were aiming for as there are too many variables measured and I do not see the links between the different variables.
RESPONSE TO THE REVIEWER. The main objective was clarified from line 78 to 81. The purpose is to evaluate how the use of eHealth in a psychonutritional programme improves people’s mental and physical well-being (psychological well-being [anxiety, depression and subjective psychological well-being] and lifestyle [overall lifestyle, diet quality and physical exercise]).
Lines 2-3. We inform of the modification of the title of the manuscript for a better understanding.
The number of participants is very small, and I wonder why the authors did not use more participants.
RESPONSE TO THE REVIEWER. This is a limitation of the study that we explain at the end of the discussion. We can remind the reviewer that the 16 participants were those who completed all the pre-, during- and post-intervention phases. We tried to use a larger sample, but because it was an online program that took place in 12 weeks, some participants dropout and some of them did not completed the last test battery (which answers were necessary to analyze the impact of ePSICONUT). We added Figure 1 to show better the evolution of the sample. Also, we added some info to justify it in lines 96 to 103. We hope that this clarify why our results are based on 16 students.
Furthermore, the authors chose to measure only trait anxiety (line 170). I wonder why they did not measure state anxiety. The experimental protocol does not include a control group, yet the authors claim that their intervention influences their dependent variables. I think the authors need to be tempered in their conclusion. As the questionnaires are self-reported, how do the authors check the veracity of the participants' answers. The choice of questionnaires used should also be justified. In sum, I think the study is not sufficiently standardised and lacks sufficient precision for readers to understand the significance of the study, although it is interesting.
RESPONSE TO THE REVIEWER. On the one hand, we evaluated both dimensions. In instruments, we describe both sections of the STAI: the state and trait section (lines 177 to 184). In results, significance values of both dimensions are reported (Table 10). On the other hand, we chose validated questionnaires/test for Spanish-speaking population to evaluate the variables of the study. Based on that, we did a baseline evaluation (before applying the program) to compare these answers with the ones that the same subjects gave to the same battery at the end of the study. Our study, as described in the article, is a pre-post intervention study, in which we analyze the impact of the program based on intra-subject answers.
The use of a control group is included as a limitation of the quasi-experimental research design. It is justified in the last paragraph of the discussion, lines 427 to 431.
Line 110: Study or studio. It has been corrected by “study”.
Table 1: People with ED. It has been corrected: eating disorder.
Lines 110-116: The acceptance number of the ethics committee must be specified. The acceptance is CIPI/19/148. It is identified in lines 106-109.
Lines 140-158: Why digital health questionnaire is different between pre and post intervention??? it's not standardised at all?
RESPONSE TO THE REVIEWER. Because the questionnaire post-intervention included an ad-hoc ePSICONUT Programme Satisfaction Survey to assess participants' perception of the following aspects of the programme: organisation, quality of the follow-up offered, quality of the information provided, the perceived impact of ePSICONUT on improving their eating habits, physical exercise and psychological well-being. Also, it includes the assessment of the specific digital tools that were used in ePSICONUT. These last questions of the digital health questionnaire couldn’t take place before the application of the programme. It has been changed and hopefully better explained in lines 149-165.
Line 147: Headspace or Haedspace. It has been corrected: Headspace.
Lines 179-191: The author could define EBP, SPSS, SPBS? Acronyms should be checked throughout the manuscript.
In Instruments section, the name of the questionnaires and what each questionnaire measures (dependent variables of the object of the present study) are indicated and hopefully better described.
Dimension of Subjective Psychological Well-Being Subscale of the Psychological Well-Being Scale (EBP). SPSS: Statistical Package for Social Sciences. Both of these acronyms have been described in the article now.
The explanations of the instruments used (questionnaires) to measure the variables under study, it has been included in 2.3.1.2. Instruments used to evaluate the psychological state and lifestyle of the participants, lines 166-223.
Lines 224-235: I suggest that the authors make a diagram to summarise the experimental protocol.
RESPONSE TO THE REVIEWER. The diagram to summarise is included in Figure 2 now (in section 2.4. Procedure and design of the intervention), lines 256-257.
Line 248: The authors say that if the data are normally distributed, they use student's t. And if not, what do they do? It is in this section that they should specify it and not in another section as the authors have done. Lines 25-35-94-95-122-125-268 and 252-455-457-… 463:
RESPONSE TO THE REVIEWER. Indeed, the statistical tests used (Wilcoxon and t-test) are described in Data Analysis, lines 268 to 282 Wilcoxon is the one that we use when the data are not normally distributed.
Choose between 'program' and 'programme' and harmonise throughout the document.
RESPONSE TO THE REVIEWER. It has been corrected: programme.
Table 3 and 5: The mixed English and Spanish authors. Care must be taken.
RESPONSE TO THE REVIEWER. Tables 3 and 5 have been corrected.
Lines 290-302: What do the observed correlations mean?
RESPONSE TO THE REVIEWER. We are grateful for the comments made. This is a conceptual error. The present study does not analyze correlations; it evaluated pre- and post-intervention differences in the variables analyzed: on the one hand, those related to lifestyle (e.g. quality of diet and physical exercise) and, on the other hand, those related to the psychological state of the participants (e.g. subjective psychological well-being, anxiety and depression). Comments about correlations have been removed.
Table 8: There is no star in the table, yet the authors have put a star at the bottom of the table to indicate a significant effect.
RESPONSE TO THE REVIEWER. We have reviewed table 8 and it does not include star at the bottom.

Reviewer 2 Report
The paper is very loosely structured and writing is not strong at all.
I give a few examples of that. Such writing mistakes are present all over the manuscript. Writing mistake examples are:
Repetition, unclear meaning, and wrong grammar:
"being perceived as powerful complementation to face to face therapies. (Line 20, 21)
Unfavorable changes in the lifestyle and in the emotional stability of individuals 39 could result on behavioral consequences that many specialists and researchers are concerned about because of the short- and long-term effects on mental and physical health (Lines 39, 40, 41)
Wrong scientific argument:
The correct (What is correct and why qualify?) use of these Information and Communication Technologies in the area health is known as eHealth. (Line 21)
Sample size is too small:
The sample con- 26 sisted of 16 university students from the Dominican Republic aged 18-24 years (x Ì… = 20.69; s = 1.74). 27 Baseline and final, pre-post intervention results were compared at the within-subjects level, obtained from psychological and lifestyle tests completed online. (Line 28)
Use of Test:
The authors have stated as following:
"To assess the impact of PSICONUT pre-intervention and post-intervention, the Wilcoxon 245 signed-rank test and Student's t-test for related samples were applied. "
Applying Wlicoxon 245 signed rank test on a sample size 16 is beyond understanding and one wonders how the results based on such a small sample size can lead give any meaningful results.
Also, age group of 18-24 (why only university students, and assertion that this group is best user of e-technologies is bit too far fetched.
Hope the authors will try to improve, data, analysis and overall writing and will resubmit their paper.
Reviewer
Author Response
We are grateful for all the considerations made for the improvement of this manuscript. We respond to each of them below.
General comments
We thank the reviewers for their feedback, which allows the authors of this manuscript to improve the application of the scientific method and scientific writing.
In general, we have modified the wording of the introduction to better relate the variables under study. In the Methodology section, the instruments used to assess the dependent variables and the protocol are explained in greater detail.
In Data analysis, we have corrected conceptual errors about the non-parametric tests used.
In Results, we tried to improve the explanation of the instruments used and the statistically significant differences found.
Finally, Discussion has been readjusted and the explanations of the limitations of the study (sample size, non-use of a control group, etc.) have been expanded.
Response to the reviewer's considerations 2
Comments to the Author
The paper is very loosely structured and writing is not strong at all.
I give a few examples of that. Such writing mistakes are present all over the manuscript. Writing mistake examples are:
- Repetition, unclear meaning, and wrong grammar: "being perceived as powerful complementation to face to face therapies. (Line 20, 21).
- Unfavorable changes in the lifestyle and in the emotional stability of individuals 39 could result on behavioral consequences that many specialists and researchers are concerned about because of the short- and long-term effects on mental and physical health (Lines 39, 40, 41).
RESPONSE TO THE REVIEWER. Thank you for the feedback. The whole text was reviewed again and checked by the authors. Also, in lines 2-3, we inform of the modification of the title of the manuscript for a better understanding and relation to the subject studied.
Wrong scientific argument:
The correct (What is correct and why qualify?) use of these Information and Communication Technologies in the area health is known as eHealth. (Line 21)
RESPONSE TO THE REVIEWER. This sentence has been changed. It was a translation mistake. (Look at line 19 and 44-45).
Sample size is too small. The sample con- 26 sisted of 16 university students from the Dominican Republic aged 18-24 years (x Ì… = 20.69; s = 1.74). 27 Baseline and final, pre-post intervention results were compared at the within-subjects level, obtained from psychological and lifestyle tests completed online. (Line 28)
RESPONSE TO THE REVIEWER. We created Figure 1 (lines 128-129) to explain better the evolution of the sample throughout the study. Also, we added a new paragraph in the article in which we discuss this limitation (lines 407-425).
Use of Test. The authors have stated as following:
"To assess the impact of PSICONUT pre-intervention and post-intervention, the Wilcoxon 245 signed-rank test and Student's t-test for related samples were applied.” Applying Wlicoxon 245 signed rank test on a sample size 16 is beyond understanding and one wonders how the results based on such a small sample size can lead give any meaningful results.
RESPONSE TO THE REVIEWER. The Wilcoxon is used to analyze differences between paired samples when the sample has a non-normal distribution (Molina & Rodrigo, 2014; T. 5 Pruebas no paramétricas. Universidad de Valencia. Retrieved from: https://fdocuments.es/document/t-5-pruebas-no-parametricas.html). This has been clarified in lines 268 a 270.
Also, age group of 18-24 (why only university students, and assertion that this group is best user of e-technologies is bit too far fetched.
RESPONSE TO THE REVIEWER. The age range is 18-45 according to the research. The 18-24 was the range of the ages of those students who completed the whole questionnaires of our study. Nevetheless, every student from 18 to 45 could originally completed them. We changed the sentence to reduce possible misunderstandings (line 94-97).

Round 2
Reviewer 1 Report
God job